# WEAKLY SUPERVISED GRAPH CONTRASTIVE LEARNING

## ABSTRACT

Graph Contrastive Learning (GCL) has recently gained popularity owing to its ability to learn efficient node representations in a self-supervised manner. These representations are typically used to train a downstream classifier. In several real-world datasets, it is difficult to acquire sufficient clean labels for classification and instead, we have weak or noisy labels available. There is little known about the robustness of the node representations learnt by the current GCL methods in the presence of weak labels. Moreover, GCL has been successfully adapted to a supervised setting where class labels are used to contrast between pairs of nodes. Can weak labels similarly be leveraged to learn better node embeddings? In this paper, we first empirically study the robustness of current GCL node representations to weak supervision. Then, we introduce Weakly Supervised Graph Contrastive Learning, WSNET, a novel method that incorporates signals from weak labels for the contrastive learning objective. We evaluate WSNET on five benchmark graph datasets comparing its performance with state-of-the-art GCL and noisy-label learning methods. We show that WSNET outperforms all baselines, particularly in the high noise setting. We conclude that although current GCL methods show great promise in the weak supervision paradigm, they are still limited in their capacity to deal with label noise and, utilizing signals from weak labels is an effective way to improve their performance.

## 1 INTRODUCTION

Despite the great success of graph neural networks (Kipf & Welling, 2016a; Veličković et al., 2017; Xu et al., 2018; Wu et al., 2020) in learning node representations, their reliance on large and clean sets of labelled data is a bottleneck (Sun et al., 2020; Zhou et al., 2019). In real-world settings, it is often challenging to procure abundant training labels to train these models. One strategy proposed to alleviate this problem is self-supervised learning (SSL) that learns graph representations without depending on training labels (Xie et al., 2022; Hu et al., 2019; Liu et al., 2021). Among SSL methods, graph contrastive learning (GCL) (Veličković et al., 2018; Hassani & Khasahmadi, 2020; You et al., 2020) typically constructs multiple "views" of an input graph via augmentations and contrasts positive pairs against negative samples to learn representations. For each given node, its positive samples are often chosen as the corresponding representations in another view, while negatives are selected from other nodes in the same view. The SSL objective has also been adapted to a supervised setup. In particular, supervised contrastive learning (SupCon) (Khosla et al., 2020b) which extends contrastive learning to a fully supervised setting has shown superior performance on the large-scale ImageNet classification task. SupCon pulls representations of nodes from the same class closer together and pushes those from different classes away from each other. Recently, the same idea was applied with success to graph contrastive learning and combined with cluster-aware data augmentation in ClusterSCL (Wang et al., 2022).

In between self-supervised and fully supervised learning, there is also weakly supervised learning, where each data point is associated with (*multiple*) noisy label(s). This is a common setup, for example, in crowdsourcing with multiple crowd workers, and when real-world datasets are labelled using a combination of rules and heuristics. Notably, the paradigm of programmatic weak supervision (Ratner et al., 2016) has proven practical in many applications (Fries et al., 2022; 2021; 2019). In this paper, we consider the scenario of weakly supervised graph contrastive learning. While there are several systematic reviews of GCL and SSL methods for graphs (Zhu et al., 2021a; Liu et al.,

2021; Xie et al., 2022; Wu et al., 2021), there are no works, to the best of our knowledge, that systematically study the robustness of the node representations learned by GCL methods to label noise. To this end, we first provide an extensive evaluation of existing GCL methods for the node classifications with noisy labels. Such a comparative study has not been done before and can reveal scope for further research in this direction.

Secondly, we explore whether weak labels help improve the quality of node representations learned using GCL. In particular, we consider the task of weakly supervised node classification, where each node has multiple noisy labels. Noisy supervision often incorrectly draws node representations of similar nodes away from each other in the embedding space. Instead of relying entirely only on the weak label of a node, we consider the distribution of weak labels of its related nodes. More clearly, we propose relying on the graph structure, specifically graph communities to find related nodes. Graph communities are relevant in several real-world graphs. For example, in social networks, individuals may be grouped together based on their interests, or in citation networks, communities may represent researchers working on closely related areas. GCL leveraging graph communities is currently under-explored. In our work, we introduce WSNET, a novel GCL method that leverages signals from both weak labels and graph communities to learn node representations.

We highlight that our setting is different from noisy label learning (NLL) where each node has one noisy label. In these cases, most solutions are focused on denoising the labels and/or loss regularization (Dai et al., 2021; Du et al., 2023). There have also been early efforts on incorporating GCL for noise robust learning (Yuan et al., 2023). In our setup, we have multiple weak labels assigned to each node, as is the norm in programmatic weak supervision.

To summarize, through this paper, we ask two main research questions. **RQ1:** How robust are the node representations learned using GCL to weakly supervised classification? **RQ2:** Can weak labels be used to learn more robust embeddings? We experiment with several GCL methods on multiple graph datasets and combine graph communities and weak labels to introduce WSNET, a new weakly supervised GCL method. We believe that answering these questions is vital in designing solutions for weakly supervised classification problems in graphs. An anonymized version of our code is released - https://anonymous.4open.science/r/StructNet-93E0/

## 2 RELATED WORK

In this section we touch upon prominent works in four related fields and we highlight how they are different from our work.

**Programmatic Weak Supervision:** In this paradigm, multiple weak supervision sources such as heuristics, knowledge bases, pre-trained models, etc are encoded into *labeling functions* (LFs). They are user-defined programs that provide labels for a subset of the data, collectively generating a large training set(Zhang et al., 2022). The LFs may be noisy, erroneous and, provide conflicting labels. To address this, label models were developed that aggregate the noisy votes of the LFs to obtain training labels which are then used to train models for downstream tasks(Ratner et al., 2016; 2019; Varma et al., 2019; Fu et al., 2020). Our work is related to PWS in that it utilizes multiple weak labels but our method is not focused on label aggregation. We simply use the signals from the weak labels to improve the contrastive learning process. We use majority vote (MV), which is the simplest and most straightforward strategy for label aggregation which chooses a label based on consensus from all the LFs. Other approaches which were designed to consider input features for a classification task either can not directly be applied to graphs, rely on pre-trained language models, or require additional inputs such as error rates of LFs or a set of labelled data (Zhang et al., 2022). We do not compete with these approaches as our focus is not on weak label aggregation or label denoising but rather studying its effects on GCL.

**Noisy Label Learning:** When there are multiple weak labels associated with each data sample (like in our problem setting), also known as programmatic weak supervision (Ratner et al., 2016; 2019), majority vote (MV) is the simplest and most straightforward strategy for label aggregation which chooses a label based on consensus from all the weak labelers. However, these MV aggregated labels are still noisy. The most common approaches for neural networks dealing with noisy labels are data-driven (Van Rooyen & Williamson, 2017), learning objective (Reed et al., 2014) or

optimization based (Arpit et al., 2017). PI-GNN (Du et al., 2023) is a recent work that introduces an adaptive noise estimation technique leveraging pairwise interactions between nodes for model regularization. NRGNN (Dai et al., 2021) is another recent work that utilizes edge prediction to predict links between unlabelled and labelled nodes and expands the training set with pseudo labels, making it more robust to label noise. Unlike these works, our method does not handle or 'clean' label noise but rather uses the weak signals to improve the node representation learning.

**Graph Contrastive Learning:**    Contrastive learning focuses on pulling a node and its *positive* sample closer to each other in the embedding space, while pushing it away from its *negative* samples (Khosla et al., 2020a; Chen et al., 2020). For contrastive learning in graphs, node and graph level augmentations are often contrasted in different ways. DGI (Veličković et al., 2018) contrasts graph and node embeddings within one augmented view. GraphCL (You et al., 2020) maximizes the agreement between two augmented views of the same graph. MVGRL (Hassani & Khasahmadi, 2020) augments the graph using node diffusion whereas GRACE (Zhu et al., 2020) augments graph views using edge removal and feature masking. CSGCL (Han et al., 2023) uses graph augmentations based on community strength and structure while GCA (Zhu et al., 2021b) uses adaptive augmentations based on topological and semantic graph properties. GAE (Kipf & Welling, 2016b) learns node embeddings by reconstructing the adjacency matrix. BGRL (Thakoor et al., 2021) predicts alternative augmentations for the nodes and alleviates the need for negative contrast pairs. GMI (Peng et al., 2020) formally generalizes mutual information for the graph domain. SUGRL (Mo et al., 2022) complements structural and neighborhood information to enlarge intra-class variation without any graph augmentations. Similarly, iGCL (Li, 2023) introduces an invariant-discriminant loss that is free from augmentations and negative samples. SelfGNN (Kefato & Girdzijauskas, 2021) proposes a GCL approach that uses feature augmentations over topological augmentations and does away with negative sampling. gCooL (Li et al., 2022) jointly learns the community partition and node representations in an end-to-end fashion showing that community information is beneficial to the overall performance. Our contrastive learning method differs from these approaches in one major way in that it leverages signals from weak labels.

**Supervised Graph Contrastive Learning:**    SupCon (Khosla et al., 2020b) introduced supervised contrastive learning for ImageNet classification and was adapted to graphs in ClusterSCL (Wang et al., 2022). To negate impacts of SupCon induced by the intra-class variances and the inter-class similarities they combine it with node clustering and cluster-aware data augmentation. JGCL (Akkas & Azad, 2022) further incorporates both supervised and self-supervised data augmentation and propose a joint contrastive loss. None of these were explicitly designed to work with one or many weak labels. (Zheng et al., 2021) proposed a weakly supervised contrastive learning framework for image classification and used node similarity to obtain weak labels. These weak labels are dependent on the augmented views of their graph and are inherently different from the noisy class labels in our setting. Moreover, they optimize a loss combination of contrasting image crops (cannot directly be applied to graphs), $L_{SupCon}$ and augmented views. We compare WSNET with the graph-adapted version of SupCon (Wang et al., 2022), which is essentially a combination of the latter two loss components. Clear (Luo et al., 2022) also relies on graph clusters to improve contrastive learning embeddings. However, ours is different in that we use graph communities for sampling positive pairs while Clear contrasts different clusters with each other for capturing underlying structural semantics. Moreover, we also leverage weak/noisy label signals while sampling positives. Recently, (Cui et al., 2023) discussed theoretically that noisy labels do not help with contrastive learning. More specifically, that it does not help select "clean labels" for training. Our work is different as firstly, we do not focus on clean label selection and secondly, our work relies on signals from *multiple* weak labels. Thirdly, our method also draws on valuable information from the graph structure which we show helps learning robust node embeddings.

## 3    PRELIMINARIES

**Graph notation:**    We represent a graph $G = (V, E)$, where $V = \{1, 2, ..., N\}$ is the set of nodes and $E$ is the set of edges. For a given node $i$ in $V$, its neighborhood $\mathcal{N}(i)$ is defined as $\{\forall j \in V | (i, j) \in E\}$. Each node $i$ is represented by a $d-$dimensional input vector $\mathcal{X}_i$ from the node feature matrix $\mathcal{X} \in \mathbb{R}^{N \times d}$ and also has an unobserved true label $y_i \in \{1, 2, ..., C\}$. $Y = [y_1, ...y_N]$ represents the true labels for all $N$ nodes in the graph.

**Weak labels:** For the setting under consideration, each node $i$ is associated with $m$ weak labels. These weak labels are assumed to be obtained from $m$ different sources also known as *labelling functions* (LF) that map an input node to a label. Each LF ($\lambda_j$) when applied on $\mathcal{X}_i$ produces a weak label $\Lambda_{ij} \in \{-1, 1, 2, \ldots, C\}$ where -1 indicates no output/abstain. $p_a$ or probability of abstain controls how many times a LF returns -1 and $p_c$ or probability of correctness, controls the accuracy of the weak labels. 1 - $p_c$ thus indicates the noise ratio in the labels. Each node $i$ is then associated with a weak label vector $\Lambda_i = \Lambda[V_i] = [\lambda_1(V_i) \ldots \lambda_m(V_i)]$ and $\Lambda \in \{-1, 1, 2, \ldots C\}^{N \times m}$ gives the weak label matrix for all $N$ nodes in the graph. Given $\Lambda_i$, an aggregated label $\tilde{y}_i$ is obtained by $MajorityVote(\Lambda_{ij})_{j=1}^m$ which is the label that most frequently appears in $\Lambda_i$. $\tilde{Y} = [\tilde{y}_1, \ldots, \tilde{y}_N]$ represents the aggregated labels for all $N$ nodes in the graph. Henceforth in this paper, *weak label* refers to $\Lambda$, *aggregated* label means $\tilde{Y}$ and $\rho$ is noise ratio.

**Problem statement:** Given $G$, $\mathcal{X}$, $\Lambda$ and $\tilde{Y}$, the goal is to learn robust node representations $H$ such that a downstream classifier model $f : H, \tilde{Y} \to Y$ can be learned.

## 4 METHODOLOGY

WSNET consists of a simple graph convolution layer parameterized by $\mathbf{W_A}$, that aggregates a node $i$'s input features ($x_i$) with that of its neighbors ($\mathcal{N}(i)$) followed by a linear layer parameterized by $\mathbf{W_L}$ and $\mathbf{B}$ to obtain hidden representations $h_i$ (Equation 1).

$$h_i = \mathcal{E}(x_i, \mathcal{N}(i)) = \mathbf{W_L}(ReLU(\mathbf{W_A} \text{AVG}(\{x_i\} \cup \{x_v | \forall v \in \mathcal{N}(i)\}))) + \mathbf{B} \tag{1}$$

These hidden representations are then mapped to the output dimension space by a fully connected linear layer followed by softmax to obtain a probability distribution over all the class labels. Drawing on ideas from information theory, WSNET optimizes a contrastive learning objective function (as per Equation 2) by maximizing the mutual information between a node and two sets of its positive samples while contrasting it with two sets of its negative samples. This contrastive loss $L$ consists of two-parts that rely on signals from the graph structure and weak node labels respectively.

$$L = L_S + L_{SupCon} \tag{2}$$

The first part $L_S$ (given by Equation 3) is based on the assumption that nodes with a similar graph structure are more likely to belong to the same class. To find a given node's positive sample, we randomly choose a node from its graph community. This forces nodes belonging to the same community to have similar representations. Extending to graph community instead of the immediate node neighborhood helps improve the quality of node representations particularly in non-homophilous networks, as confirmed by our experiments. Additionally, we also consider the similarity between the weak label distributions of nodes while selecting positives and negatives. More clearly, for a given node, we calculate the dot product similarity between the frequency vector of its weak labels and those of all other nodes belonging to its community and pick the node with the highest similarity. Likewise, the negatives are sampled from outside of the community based on their dissimilarity with the given node's weak label frequency vector. We used the popular Louvain algorithm for finding graph communities. This process is outlined in Algorithm 1.

$$L_S = -\sum_{i=1}^{|V|} \log \frac{\exp\left(h_i.h_i^+/\tau\right)}{\sum_{j \in \tilde{K}_i} \exp\left(h_i.h_j/\tau\right) + \exp\left(h_i.h_i^+/\tau\right)} \tag{3}$$

Here $h_i^+$ is the positive sample for node $i$, $\tilde{K}_i$ is the set of its $r$ negative samples and $\tau$ is the temperature parameter. '.' indicates dot product.

The second part $L_{SupCon}$ (given by Equation 4) is the SupCon loss function. Here, we sample a node's positive from the set of nodes that has the same aggregated label and negatives from the remaining nodes. Typically, SupCon is used in a semi-supervised setting with a small percentage of labelled nodes which limits its performance. In our setting, since weak labels are cheap, we can obtain them for all nodes, thus providing some label information for all the nodes and improving the effect of using the SupCon loss.

$$L_{SupCon} = -\sum_{i=1}^{|V|} \sum_{p \in P_i} \log \frac{\exp\left(h_i.h_p/\tau\right)}{\sum_{n \in K_i} \exp\left(h_i.h_n/\tau\right) + \sum_{p \in P_i} \exp\left(h_i.h_p/\tau\right)} \tag{4}$$

$P_i$ is the set of nodes with the same aggregated class label as node $i$ and $h_p$ indicates the corresponding node representations. $K_i$ indicates the $r$ negative samples and $\tau$ is the temperature parameter. The procedure for obtaining $P_i$ and $K_i$ is outlined in lines 15 and 16 in Algorithm 2.

The loss function $L$ is optimized to learn the model parameters and obtain embeddings that can then be fed into a downstream classifier such as a logistic regression model and trained using the aggregated labels. Algorithm 2 details the steps involved in WSNET.

---

**Algorithm 1** FINDCONTRASTPAIRS

**Input**: $\Lambda$, $Comms$, $V_i$, $V$, $r$
**Output**: $V_i^+$, $V_i^-$

1:   $ll \leftarrow \Lambda[V_i]$
2:   $f_i \leftarrow frequncyCounter(ll)$                        ▷ counts the frequency of each class in $ll$
3:   $C_i \leftarrow Comms[V_i]$
4:   $NC_i \leftarrow V \setminus Comms[V_i]$
5:   $prob^+ \leftarrow emptyList(size = |C_i|)$
6:   $prob^- \leftarrow emptyList(size = |NC_i|)$
7:   **for** $j \in C_i$ **do**
8:      $\lambda_j \leftarrow \Lambda[j]$
9:      $f_j \leftarrow frequncyCounter(\lambda_j)$
10:     $prob^+[j] \leftarrow f_i \cdot f_j$                            ▷ cosine similarity
11: **end for**
12: $V_i^+ \sim randomChoice(C_i, prob^+)$            ▷ sample from $prob^+$
13: **for** $j \in NC_i$ **do**
14:     $\lambda_j \leftarrow \Lambda[j]$
15:     $f_j \leftarrow frequncyCounter(\lambda_j)$
16:     $prob^-[j] \leftarrow 1 - (f_i \cdot f_j)$                 ▷ cosine dissimilarity
17: **end for**
18: $V_i^- \leftarrow emptyList(size = r)$
19: **for** $k \in [1 \ldots r]$ **do**
20:     $V_i^-[k] \leftarrow v^- \sim randomChoice(NC_i, prob^-)$      ▷ sample from $prob^-$
21: **end for**
22: **return** $V_i^+$, $V_i^-$

---

## 5 EXPERIMENTS

We ran two sets of experiments to address **RQ1** and **RQ2** described as follows. First, we evaluated the node embeddings learned by various GCL methods on their robustness to aggregated labels in a downstream node classification task. We compared across different styles of contrastive learning with and without augmented views, negative sampling and neighbourhood based positive sampling with the goal of studying any correlation with their performance on weakly supervised classification. Second, we evaluated WSNET for the same task and compared it to relevant baselines, including self-supervised learning (SSL), noisy label learning (NLL) and supervised GCL (Sup-GCL).

For both experiments, we synthetically created weak labels using Algorithm 3 for fixed coverage ($p_c$), while varying accuracy ($p_a$) and number of weak labels ($m$). We use a simple majority vote strategy to aggregate the $m$ weak labels to get a single aggregated label for each node. The values of $(m, p_a)$ were arbitrarily chosen [(5, 0.45), (10, 0.65), (50, 0.55)] to ensure that the aggregated labels have accuracy of around $47\%, 68\%$ and $90\%$, i.e, having a label noise rate of $53\%, 32\%$ and $10\%$ respectively. These values were chosen to correspond to a high, medium and low noise setting for evaluating our method. $p_a$ was fixed at $0.2$ to simulate a difficult setup where $80\%$ of the time, each of the LFs ($\lambda_j$) returned $-1$.

We repeated each experiment 5-times on different 80-20 train-test splits and reported the average and standard deviation across all the runs. For each run, both the GCL component and the downstream classifier had access to weak labels only from the train split during training. For all baselines, we used their official code base available online and retained all the default hyperparameter settings. For WSNET, we set $r$ to be 10 and $\tau$ as 0.65 based on grid-search and $T$ (training epochs) to be 300.

---

**Algorithm 2** WSNET

---
**Input**: $\mathcal{X}, \Lambda, V, E, T$
**Hyperparameters**: $r, \tau$
**Output**: $H$

1:  $t \leftarrow 1$
2:  $Comms \leftarrow Louvain(V, E, \mathcal{X})$       ▷ Finding communities using (Blondel et al., 2008)
3:  $Labels \leftarrow MajorityVote(\Lambda)$
4:  **while** $t \leq T$ **do**
5:    **for** $x_i \in \mathcal{X}$ **do**
6:     $h_i \leftarrow \mathcal{E}(x_i, \mathcal{N}(i))$            ▷ Using Equation 1
7:     $V^+, V^- \leftarrow$ FINDCONTRASTPAIRS$(\Lambda, Comms, V_i, V, r)$   ▷ Using Algorithm 1
8:     $h_i^+ \leftarrow \mathcal{E}(x_{V^+}, \mathcal{N}(V^+))$
9:     $\tilde{K}_i \leftarrow emptyList(size = r)$
10:    **for** $j \in [1 \ldots r]$ **do**
11:     $\tilde{K}_i[j] \leftarrow \mathcal{E}(x_{V_j^-}, \mathcal{N}(V_j^-))$
12:    **end for**
13:   **end for**
14:   Calculate $L_S$ using Equation 3
15:   $P_i \leftarrow V_k : Labels[V_k] = Labels[V_i]$
16:   $K_i \leftarrow V_k : Labels[V_k] \neq Labels[V_i]$
17:   Calculate $L_{SupCon}$ using Equation 4
18:   $L = L_S + L_{SupCon}$
19:   Backpropagate $L$ and update all parameter weights
20:   $t \leftarrow t + 1$
21: **end while**
22: $H \leftarrow [\mathcal{E}(\mathcal{X}_i, \mathcal{N}(i)) | \forall i \in V]$
23: **Return**: $H$

---

For all GCL methods, the learned embeddings were used to train a downstream Logistic Regression classifier unless specified otherwise and the classification performance on the test set was reported. All the experiments were run on a local 8-cpu core Macbook M2 computer.

---

**Algorithm 3** SYNTHETICLABELING

---
**Input**: $y, C, p_c, p_a, m$
**Output**: $\hat{\lambda}$

1:  $L \leftarrow [0, 1 \ldots C, -1]$
2:  $\hat{\lambda} \leftarrow emptylist(size = m)$
3:  **for** $i \in [1 \ldots m]$ **do**
4:    $P \leftarrow emptylist(size = C + 1)$
5:    $P[-1] \leftarrow p_a$            ▷ prob. of abstain
6:    $P[y] \leftarrow (1 - p_a) \times p_c$       ▷ prob. of voting $\times$ accuracy
7:    **for** $l \in L \setminus \{-1, y\}$ **do**
8:     $P[l] \leftarrow ((1 - p_a) \times (1 - p_c))/(C - 1)$
9:    **end for**
10:   $\hat{\lambda}[i] \sim randomChoice(L, P)$    ▷ sample a label at random with probability $P$
11: **end for**
12: **Return**: $\hat{\lambda}$

---

**Datasets:** We run our experiments on five benchmark node classification datasets, namely Cora, Citeseer, Pubmed, Texas and Wisconsin. The first three are citation networks where the nodes are academic papers, edges indicate if a paper was cited/cites another paper and classes are topics that the papers belong to. Texas and Wisconsin are college webpage datasets where the nodes represent webpages, edges are hyperlinks between them and the nodes may be classified as student, staff, etc[1].

---

[1]Texas and Wisconsin were taken from Non-Homophily-Large-Scale. All other datasets are from Deep Graph Library.

Cora, Citeseer and Pubmed are homophilous whereas Texas and Wisconsin as non-homophilous. Algorithm 3 was used to generate the weak labels for all the datasets. These graphs also vary in their properties such as number of nodes, edges, classes and homophily as summarized in Table 1. Due to its large size, while selecting negative pairs for Pubmed, only 1000 out-of-community nodes, chosen at random, were considered.

Table 1: Dataset statistics. $|V|$, $|E|$ and $C$ are the number of nodes, edges and classes in the graph respectively. Homophily shows to what degree neighboring nodes have the same label and Communities represents the number of groups found by the Louvain algorithm.

| Dataset | $|V|$ | $|E|$ | $C$ | Homophily | Communities |
|---------|-------|-------|-----|-----------|-------------|
| Cora | 2995 | 8158 | 7 | 0.83 | 81 |
| Citeseer | 4230 | 5337 | 6 | 0.71 | 537 |
| Pubmed | 19717 | 44324 | 3 | 0.79 | 39 |
| Texas | 183 | 350 | 5 | 0.11 | 62 |
| Wisconsin | 251 | 482 | 5 | 0.21 | 88 |

**GCL methods:** For **RQ1**, we compare the performance of several GCL baselines highlighted in Table 4. We include GCL methods that rely on using augmented views of nodes, contrasting with negative samples, sampling positives from node neighborhoods or some combination of these.

WSNET falls under the category of *weakly supervised graph contrastive learning* (WS-GCL) and for **RQ2**, we compare it with baselines from related areas namely self-supervised learning (SSL) or GCL, noisy label learning (NLL) and supervised graph contrastive learning (SupGCL). For SSL, we use the same baselines from Table 4. NLL baselines include recent works NRGNN (Dai et al., 2021) and PI-GNN (Du et al., 2023) both of which explicitly deal with noisy labels while performing node classification. From Sup-GCL, we include SupCon (Khosla et al., 2020a), Joint Training of an augmented view contrastive loss and SupCon similar to (Akkas & Azad, 2022) and included in (Cui et al., 2023) and ClusterSCL (Wang et al., 2022) that learns cluster assignments for nodes along with supervised GCL.

**RQ1 Results - Robustness of GCL to label noise:** The results for the first set of experiments are in Table 2. The values in the table indicate the weighted F1 classification score and the values within brackets show the percentage decrease in weighed F1 due to label noise. i.e, the same embeddings are used to train two logistic regression models with the true labels and weak labels respectively and their difference is tabulated to measure their *robustness* to noise.

In the high and medium label noise settings (Tables 2a and 2b), we see that the performance of almost all baselines are very close and there is no clear winner across all datasets. The three baselines that use neighbourhood-based sampling of positive pairs do better than others in the high noise setting. For example, see SUGRL (Mo et al., 2022) on Citeseer and iGCL (Li, 2023) on Wisconsin from Table 2a. Such neighborhood based sampling that contrasts views of neighboring nodes instead of augmented views of the same nodes, allows the learned embeddings to contain even more information about the neighbors, contributing to their robustness. When the noise ratio is low (Table 2c), MVGRL (Hassani & Khasahmadi, 2020) and GraphCL (You et al., 2020) are best performing in the homophilous datasets (Cora, Citeseer and Pubmed). However, no clear recommendations can be made as to the effect of augmentations or negative sampling on the classification performance. These results provide reasonable evidence that there is a need for methods specifically focusing on weakly supervised classification.

**RQ2 Results - Performance of WSNET:** Table 3 presents the weighted F1 classification score and results of the second experiment. We carry over the best performing result from Table 2 to compare it with the remaining baselines. WSNET + LR uses the embeddings learned by our method to train a logistic regression model for classification and WSNET + RF uses a random forest classifier. We included the RF downstream classifier to make it a fairer competitor to the NLL baselines as it is known to be more robust to label noise compared to logistic regression (Ishii & Ljunggren, 2021). WSNET + LR is more comparable to all the GCL baselines including Sup-GCL. First, we note that WSNET + LR outperforms all SSL and Sup-GCL methods on Cora, Citeseer, Texas and Wisconsin

Table 2: Robustness comparison of GCL methods to label noise. We compare the classification performance (weighted F1 score) of different GCL methods in three noise settings: low, medium, and high. In parenthesis, we report the percentage decrease in performance compared to when the method has access to the true labels. We observe that the performance of all methods degrades as label noise increases and there is no single method that clearly outperforms others across different noise settings.

| | Cora | Citeseer | Pubmed | Texas | Wisconsin |
|---|---|---|---|---|---|
| DGI (Veličković et al., 2018) | 0.26 (70.1%) | 0.29 (67.6%) | 0.37 (57.0%) | 0.25 (38.8%) | 0.45 (18.8%) |
| GAE (Kipf & Welling, 2016b) | 0.23 (28.5%) | 0.29 (34.1%) | 0.42 (49.5%) | 0.23 (54.6%) | 0.38 (15.0%) |
| CSGCL (Han et al., 2023) | 0.32 (59.3%) | 0.31 (63.0%) | 0.41 (49.8%) | 0.15 (62.8%) | 0.35 (6.40%) |
| GRACE (Zhu et al., 2020) | 0.32 (58.4%) | 0.33 (60.8%) | 0.41 (51.6%) | 0.26 (37.5%) | 0.30 (12.7%) |
| MVGRL (Hassani & Khasahmadi, 2020) | 0.33 (60.9%) | 0.32 (64.7%) | 0.40 (53.9%) | 0.28 (25.9%) | 0.37 (5.32%) |
| GraphCL (You et al., 2020) | 0.32 (62.6%) | 0.26 (70.5%) | 0.38 (56.4%) | 0.16 (53.0%) | 0.32 (31.2%) |
| GCA (Zhu et al., 2021b) | **0.34** (58.9%) | 0.31 (63.6%) | 0.42 (50.2%) | 0.28 (22.5%) | 0.24 (38.0%) |
| BGRL (Thakoor et al., 2021) | 0.33 (48.4%) | 0.31 (49.6%) | **0.44** (38.9%) | 0.26 (21.5%) | 0.37 (8.53%) |
| GMI (Peng et al., 2020) | 0.31 (49.3%) | 0.34 (53.6%) | 0.37 (48.3%) | 0.29 (15.1%) | 0.42 (8.13%) |
| SUGRL (Mo et al., 2022) | 0.32 (51.3%) | **0.38** (47.2%) | 0.39 (45.9%) | 0.30 (15.7%) | 0.33 (24.6%) |
| iGCL (Li, 2023) | 0.32 (49.8%) | 0.28 (58.3%) | 0.41 (44.8%) | 0.21 (21.0%) | **0.45** (0.63%) |
| SelfGNN (Kefato & Girdzijauskas, 2021) | 0.24 (55.5%) | 0.28 (56.3%) | 0.42 (43.3%) | **0.32** (7.94%) | 0.41 (15.4%) |

(a) High noise setting: $\rho = 53\%$

| | Cora | Citeseer | Pubmed | Texas | Wisconsin |
|---|---|---|---|---|---|
| DGI (Veličković et al., 2018) | 0.53 (35.8%) | 0.55 (38.1%) | **0.62** (28.9%) | 0.20 (64.6%) | 0.38 (25.1%) |
| GAE (Kipf & Welling, 2016b) | 0.46 (24.5%) | 0.51 (21.9%) | **0.62** (17.3%) | 0.21 (51.5%) | 0.33 (41.9%) |
| CSGCL (Han et al., 2023) | 0.48 (38.8%) | 0.53 (37.1%) | 0.58 (28.7%) | 0.22 (48.6%) | 0.28 (10.9%) |
| GRACE (Zhu et al., 2020) | 0.46 (39.5%) | 0.53 (37.6%) | 0.60 (28.2%) | 0.16 (61.6%) | 0.34 (10.5%) |
| MVGRL (Hassani & Khasahmadi, 2020) | 0.57 (34.2%) | **0.60** (32.8%) | 0.61 (28.9%) | 0.21 (45.5%) | 0.36 (18.0%) |
| GraphCL (You et al., 2020) | **0.58** (33.1%) | 0.57 (34.0%) | 0.61 (30.2%) | 0.12 (75.2%) | 0.36 (36.0%) |
| GCA (Zhu et al., 2021b) | 0.54 (33.6%) | 0.51 (39.2%) | 0.58 (29.5%) | 0.29 (22.4%) | 0.33 (25.4%) |
| BGRL (Thakoor et al., 2021) | 0.48 (33.2%) | 0.47 (35.8%) | 0.57 (25.5%) | **0.30** (26.1%) | 0.38 (5.88%) |
| GMI (Peng et al., 2020) | 0.53 (28.5%) | 0.56 (32.1%) | 0.60 (25.9%) | 0.17 (48.5%) | 0.36 (13.7%) |
| SUGRL (Mo et al., 2022) | 0.57 (27.3%) | 0.58 (27.3%) | 0.60 (24.5%) | 0.22 (29.6%) | **0.47** (15.2%) |
| iGCL (Li, 2023) | 0.49 (32.9%) | 0.53 (31.7%) | 0.59 (26.9%) | 0.13 (29.5%) | 0.35 (7.36%) |
| SelfGNN (Kefato & Girdzijauskas, 2021) | 0.51 (31.5%) | 0.53 (29.9%) | 0.60 (25.6%) | 0.13 (15.9%) | 0.34 (9.10%) |

(b) Medium noise setting: $\rho = 32\%$

| | Cora | Citeseer | Pubmed | Texas | Wisconsin |
|---|---|---|---|---|---|
| DGI (Veličković et al., 2018) | 0.79 (6.73%) | 0.80 (7.83%) | **0.72** (17.5%) | 0.35 (11.2%) | 0.39 (25.9%) |
| GAE (Kipf & Welling, 2016b) | 0.77 (10.7%) | 0.72 (7.37%) | 0.70 (16.3%) | 0.49 (15.1%) | 0.43 (8.54%) |
| CSGCL (Han et al., 2023) | 0.72 (7.25%) | 0.78 (8.26%) | 0.69 (17.3%) | 0.37 (22.1%) | 0.37 (6.38%) |
| GRACE (Zhu et al., 2020) | 0.70 (10.7%) | 0.79 (7.17%) | 0.71 (16.7%) | 0.33 (10.2%) | 0.34 (4.59%) |
| MVGRL (Hassani & Khasahmadi, 2020) | **0.81** (6.49%) | **0.81** (9.42%) | **0.72** (16.5%) | 0.48 (0.73%) | 0.36 (5.69%) |
| GraphCL (You et al., 2020) | 0.80 (5.43%) | **0.81** (7.35%) | **0.72** (17.2%) | 0.43 (10.5%) | 0.32 (32.3%) |
| GCA (Zhu et al., 2021b) | 0.74 (8.50%) | 0.77 (8.36%) | 0.69 (17.3%) | 0.49 (4.85%) | 0.34 (18.5%) |
| BGRL (Thakoor et al., 2021) | 0.73 (7.67%) | 0.72 (10.3%) | 0.69 (14.4%) | **0.53** (5.57%) | 0.39 (0.45%) |
| GMI (Peng et al., 2020) | 0.76 (5.04%) | 0.79 (7.04%) | **0.72** (14.4%) | 0.44 (10.6%) | **0.47** (2.65%) |
| SUGRL (Mo et al., 2022) | 0.77 (6.60%) | 0.78 (6.87%) | 0.71 (14.2%) | 0.43 (2.60%) | 0.39 (12.8%) |
| iGCL (Li, 2023) | 0.76 (5.98%) | 0.77 (9.25%) | 0.71 (15.5%) | 0.42 (1.63%) | 0.42 (2.15%) |
| SelfGNN (Kefato & Girdzijauskas, 2021) | 0.74 (6.71%) | 0.76 (7.77%) | 0.71 (14.4%) | 0.42 (1.80%) | 0.39 (11.5%) |

(c) Low noise setting: $\rho = 10\%$

for all noise settings. On Cora, Texas and Wisconsin, it also beats the NLL baselines. WSNET + RF further outperforms all baselines particularly in the high noise setting. When the noise ratio is medium and low, WSNET performs best on Cora, Citeseer, Texas and Wisconsin. It is particularly worth noting that WSNET + RF outperforms noisy label learning methods like NRGNN and PI-GNN. These NLL baselines are strong contenders on the homophilous graphs, especially Pubmed, but do not perform as well on Texas and Wisconsin. We highlight that WSNET particularly works very well on these non-homophilous graphs which we believe can be attributed to our community-based sampling approach. Texas is a small dataset with only 183 nodes, and although on average it seems that WSNET's performance on high noise setting is better compared to medium noise, we observe a high variance in the results resulting in an overlap in the ranges. This is also only observed using LR classifier and not RF. Thus, more than the quality of the WSNET embeddings, this behaviour may party be attributed to the simplicity of the downstream LR classifier.

We also include two ablations of our method where either part of the two-part loss function is removed, indicated by $-L_S$ and $-L_{SupCon}$. We see that our method combining the two is best performing especially in the high noise setting. Additionally, WSNET facilitates the pre-computation of the positive and negative samples for all nodes as they only depend on the graph structure and weak labels and not on the learned embeddings. Moreover, on Pubmed, the negative pairs of nodes were selected based on weak label dissimilarity from 1000 randomly chosen out-of-community nodes,

without practically harming the performance. Thus, WSNET can also be adapted to be scaled to large graphs.

Table 3: Performance of the proposed WSNET compared to baselines of different type: SSL (Self-Supervised Learning), NLL (Noisy Label Learning), and Sup-GCL (Supervised Graph Contrastive Learning). Our method is the first proposal for a Weakly Supervised Graph Contrastive Learning (WS-GCL) approach. The results are weighted F1 classification score averaged across 5 train-test splits. We can see that WSNET has a clear advantage in the high noise setting and overall performs better than all the baselines. The bold and overline show the best and second best F1 scores respectively for each dataset.

| Method | Cora | Citeseer | Pubmed | Texas | Wisconsin | Type |
|---|---|---|---|---|---|---|
| Best from Table 2 | $0.34 \pm 0.20$ | $0.38 \pm 0.03$ | $0.44 \pm 0.14$ | $0.32 \pm 0.01$ | $0.45 \pm 0.19$ | SSL |
| NRGNN (Dai et al., 2021) | $0.21 \pm 0.23$ | $0.18 \pm 0.35$ | $0.24 \pm 0.02$ | $0.09 \pm 0.08$ | $0.31 \pm 0.04$ | NLL |
| PI-GNN (Du et al., 2023) | $0.65 \pm 0.04$ | $\overline{0.61} \pm 0.29$ | $\overline{0.50} \pm 0.09$ | $0.13 \pm 0.13$ | $0.35 \pm 0.10$ | |
| SupCon (Khosla et al., 2020b) | $0.48 \pm 0.14$ | $0.54 \pm 0.30$ | $0.34 \pm 0.03$ | $0.43 \pm 0.04$ | $0.35 \pm 0.10$ | |
| Joint Training (Cui et al., 2023) | $0.48 \pm 0.15$ | $0.47 \pm 0.19$ | $0.28 \pm 0.06$ | $0.46 \pm 0.05$ | $0.35 \pm 0.03$ | Sup-GCL |
| ClusterSCL (Wang et al., 2022) | $0.35 \pm 0.06$ | $0.49 \pm 0.11$ | $0.39 \pm 0.04$ | $0.30 \pm 0.05$ | $0.37 \pm 0.09$ | |
| WSNET- $L_{SupCon}$ | $0.44 \pm 0.07$ | $0.49 \pm 0.11$ | $0.40 \pm 0.01$ | $0.41 \pm 0.10$ | $0.56 \pm 0.10$ | |
| WSNET- $L_S$ | $0.67 \pm 0.04$ | $0.55 \pm 0.05$ | $0.32 \pm 0.01$ | $\overline{0.53} \pm 0.07$ | $0.50 \pm 0.09$ | WS-GCL |
| WSNET + LR | $\mathbf{0.68} \pm 0.02$ | $0.54 \pm 0.05$ | $0.41 \pm 0.01$ | $\mathbf{0.68} \pm 0.09$ | $\overline{0.59} \pm 0.07$ | (ours) |
| WSNET + RF | $\overline{0.67} \pm 0.08$ | $\mathbf{0.85} \pm 0.16$ | $\mathbf{0.60} \pm 0.01$ | $0.41 \pm 0.04$ | $\mathbf{0.62} \pm 0.09$ | |

(a) High noise setting: $\rho = 53\%$

| Method | Cora | Citeseer | Pubmed | Texas | Wisconsin | Type |
|---|---|---|---|---|---|---|
| Best from Table 2 | $0.58 \pm 0.22$ | $0.60 \pm 0.31$ | $0.62 \pm 0.06$ | $0.30 \pm 0.03$ | $0.47 \pm 0.04$ | SSL |
| NRGNN (Dai et al., 2021) | $0.81 \pm 0.05$ | $0.23 \pm 0.14$ | $0.75 \pm 0.12$ | $0.17 \pm 0.15$ | $0.40 \pm 0.10$ | NLL |
| PI-GNN (Du et al., 2023) | $0.77 \pm 0.07$ | $\overline{0.69} \pm 0.34$ | $\mathbf{0.86} \pm 0.00$ | $0.43 \pm 0.05$ | $0.43 \pm 0.08$ | |
| SupCon (Khosla et al., 2020b) | $0.48 \pm 0.12$ | $0.65 \pm 0.22$ | $0.38 \pm 0.05$ | $0.45 \pm 0.03$ | $0.36 \pm 0.03$ | |
| Joint Training (Cui et al., 2023) | $0.49 \pm 0.14$ | $0.48 \pm 0.27$ | $0.35 \pm 0.04$ | $0.48 \pm 0.04$ | $0.35 \pm 0.04$ | Sup-GCL |
| ClusterSCL (Wang et al., 2022) | $0.27 \pm 0.07$ | $0.59 \pm 0.19$ | $0.29 \pm 0.07$ | $0.23 \pm 0.03$ | $0.37 \pm 0.04$ | |
| WSNET- $L_{SupCon}$ | $0.50 \pm 0.01$ | $0.52 \pm 0.16$ | $0.59 \pm 0.01$ | $0.31 \pm 0.04$ | $0.54 \pm 0.08$ | |
| WSNET- $L_S$ | $\overline{0.81} \pm 0.01$ | $0.59 \pm 0.06$ | $0.62 \pm 0.01$ | $0.49 \pm 0.07$ | $0.55 \pm 0.06$ | WS-GCL |
| WSNET + LR | $0.80 \pm 0.04$ | $0.65 \pm 0.10$ | $0.59 \pm 0.01$ | $\overline{0.56} \pm 0.08$ | $\overline{0.63} \pm 0.08$ | (ours) |
| WSNET + RF | $\mathbf{0.83} \pm 0.03$ | $\mathbf{0.87} \pm 0.14$ | $\overline{0.76} \pm 0.01$ | $\mathbf{0.58} \pm 0.05$ | $\mathbf{0.66} \pm 0.07$ | |

(b) Medium noise setting: $\rho = 32\%$

| Method | Cora | Citeseer | Pubmed | Texas | Wisconsin | Type |
|---|---|---|---|---|---|---|
| Best from Table 2 | $0.81 \pm 0.13$ | $0.81 \pm 0.05$ | $0.72 \pm 0.21$ | $0.53 \pm 0.02$ | $0.47 \pm 0.16$ | SSL |
| NRGNN (Dai et al., 2021) | $0.75 \pm 0.12$ | $0.44 \pm 0.13$ | $0.82 \pm 0.01$ | $0.42 \pm 0.03$ | $0.35 \pm 0.13$ | NLL |
| PI-GNN (Du et al., 2023) | $0.83 \pm 0.05$ | $\mathbf{0.89} \pm 0.04$ | $\mathbf{0.87} \pm 0.01$ | $0.53 \pm 0.01$ | $0.43 \pm 0.08$ | |
| SupCon (Khosla et al., 2020b) | $0.35 \pm 0.15$ | $0.65 \pm 0.21$ | $0.35 \pm 0.07$ | $0.50 \pm 0.06$ | $0.38 \pm 0.07$ | |
| Joint Training (Cui et al., 2023) | $0.27 \pm 0.13$ | $0.49 \pm 0.30$ | $0.30 \pm 0.06$ | $0.49 \pm 0.02$ | $0.38 \pm 0.04$ | Sup-GCL |
| ClusterSCL (Wang et al., 2022) | $0.44 \pm 0.09$ | $0.75 \pm 0.25$ | $0.35 \pm 0.08$ | $0.45 \pm 0.03$ | $0.37 \pm 0.06$ | |
| WSNET- $L_{SupCon}$ | $0.79 \pm 0.07$ | $0.76 \pm 0.08$ | $0.70 \pm 0.01$ | $0.52 \pm 0.05$ | $0.53 \pm 0.06$ | |
| WSNET- $L_S$ | $\overline{0.94} \pm 0.01$ | $0.58 \pm 0.03$ | $0.71 \pm 0.01$ | $0.68 \pm 0.04$ | $0.61 \pm 0.08$ | WS-GCL |
| WSNET + LR | $0.92 \pm 0.02$ | $\overline{0.85} \pm 0.09$ | $0.70 \pm 0.01$ | $\overline{0.72} \pm 0.06$ | $\overline{0.75} \pm 0.07$ | (ours) |
| WSNET + RF | $\mathbf{0.95} \pm 0.01$ | $\mathbf{0.89} \pm 0.03$ | $0.81 \pm 0.01$ | $\mathbf{0.77} \pm 0.08$ | $\mathbf{0.83} \pm 0.09$ | |

(c) Low noise setting: $\rho = 10\%$

# 6  CONCLUSION

In this paper, we systematically explored the robustness of GCL methods to label noise and piloted studies on weakly supervised graph contrastive learning. We found that current GCL and supervised GCL solutions can be made more robust to weak labels and to this end proposed WSNET that leverages signals from the graph community structure as well as weak labels. WSNET improves upon unsupervised and supervised GCL as well as noisy label learning methods on 5 graph datasets, especially on non-homophilous graphs. Despite its effectiveness, our method is constrained by the quality of the community detection algorithm used and one future direction of research includes improving on them. Another direction is to explore weakly supervised GCL as a regularization term in noisy label learning methods.

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

## A GCL METHODS

We provide a summary of the different GCL methods compared in our work in Table 4. We also indicate whether a GCL method includes augmentation, negative sampling, neighborhood-based sampling, supervisory signals or weak labels.

## B SENSITIVITY TO PARAMETER $r$

$r$ indicates the number of negative pairs sampled for the contrastive loss. Generally, the higher the value of $r$, the better the performance. However, with higher values of $r$, the computational complexity also increases as the denominator term in the $L_S$ (Equation 3). We ran experiments on the Cora dataset for varying values of $r$ and reported the results in Table 5 below. We observe a similar performance for $r = 10$ and $r = 20$ with slight increase in runtime. Given that the performance is similar, we stick to $r = 10$ as in our experiments in the paper.

| Method | Augmentation | Negatives | Neighbourhood-based | Supervision | Weak Labels |
|---|---|---|---|---|---|
| DGI (Veličković et al., 2018) | ✗ | ✓ | ✗ | ✗ | ✗ |
| CSGCL (Han et al., 2023) | ✓ | ✓ | ✗ | ✗ | ✗ |
| GRACE (Zhu et al., 2020) | ✓ | ✓ | ✗ | ✗ | ✗ |
| MVGRL (Hassani & Khasahmadi, 2020) | ✓ | ✓ | ✗ | ✗ | ✗ |
| GraphCL (You et al., 2020) | ✓ | ✓ | ✗ | ✗ | ✗ |
| GCA (Zhu et al., 2021b) | ✓ | ✓ | ✗ | ✗ | ✗ |
| BGRL (Thakoor et al., 2021) | ✓ | ✗ | ✗ | ✗ | ✗ |
| GMI (Peng et al., 2020) | ✗ | ✓ | ✓ | ✗ | ✗ |
| SUGRL (Mo et al., 2022) | ✗ | ✓ | ✓ | ✗ | ✗ |
| iGCL (Li, 2023) | ✗ | ✗ | ✓ | ✗ | ✗ |
| SelfGNN (Kefato & Girdzijauskas, 2021) | ✓ | ✗ | ✗ | ✗ | ✗ |
| SupCon (Khosla et al., 2020b) | ✓ | ✓ | ✗ | ✓ | ✗ |
| Joint Training (Cui et al., 2023) | ✓ | ✓ | ✗ | ✓ | ✗ |
| ClusterSCL (Wang et al., 2022) | ✓ | ✓ | ✗ | ✓ | ✗ |
| WSNET (Ours) | ✗ | ✓ | ✓ | ✗ | ✓ |

Table 4: Summary of different GCL methods. Our method does not use augmentations and is designed to work with weak labels. ✓ indicates presence of that property and ✗ indicates absence.

| Cora | 53% | 32% | 10% | Runtime (s) |
|---|---|---|---|---|
| $r = 5$ | $0.40 \pm 0.04$ | $0.74 \pm 0.07$ | $0.93 \pm 0.05$ | 25.4 |
| $r = 10$ | $0.68 \pm 0.02$ | $0.80 \pm 0.04$ | $0.92 \pm 0.02$ | 28.3 |
| $r = 20$ | $0.68 \pm 0.04$ | $0.80 \pm 0.06$ | $0.93 \pm 0.07$ | 28.6 |

Table 5: F1 classification score on Cora dataset for varying noise levels and different values of $r$.

## C  TRAINING DETAILS AND RUNTIME COMPLEXITY

All the experiments were run locally on an 8-cpu core Macbook M2 computer. WSNet was trained for 50 epochs and the results averaged over 5 runs is reported in the paper. For all the other GCL methods, we used the official code released by the authors of the respective papers and tried our best to use the recommended hyperparameters wherever relevant.

Community detection and identifying both positives and negatives has a total runtime complexity of $O(nlogn) + O(np^2)$ where $n$ is the number of nodes in the graph and $p$ is the size of the largest community in the graph and $p << n$.

## D  VISUALIZATION OF LEARNED EMBEDDINGS

We plot the TSNE representations of the embeddings learned by WSNET on Cora dataset and baselines including ClusterSCL, SelfGNN, iGCL, SUGRL and BGRL (before the classification component) in Figure 1. The colors in the plot represent the class labels (7 classes in Cora). We observe that WSNET embeddings are most separabale with respect to class labels compared to other GCL methods despite being affected by the noise in the weak labels. Apart from WSNET and Cluster-SCL, the other GCL method do not use any label information and hence are not affected by the noise in the labels. Compared to ClusterSCL which is also cluster-based and using the noisy weak labels for contrastive learning, we show that WSNET achieves better class separability.

Additionally, we further study the embeddings of WSNET, ClusterSCL and SUGRL by training a linear classifier (Logistic Regression) that we used in our experiments. We also plotted the TSNE representations of the predicted probability vectors under different levels of noise in Figure 2.

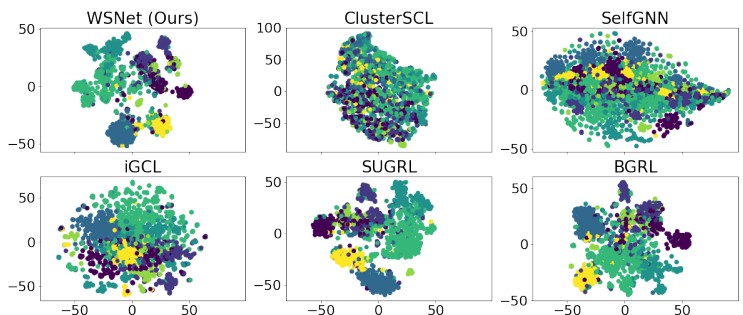

Figure 1: TSNE representations of embeddings learned by various methods on Cora dataset. The colors indicate class labels. WSNET shows most separability of classes.

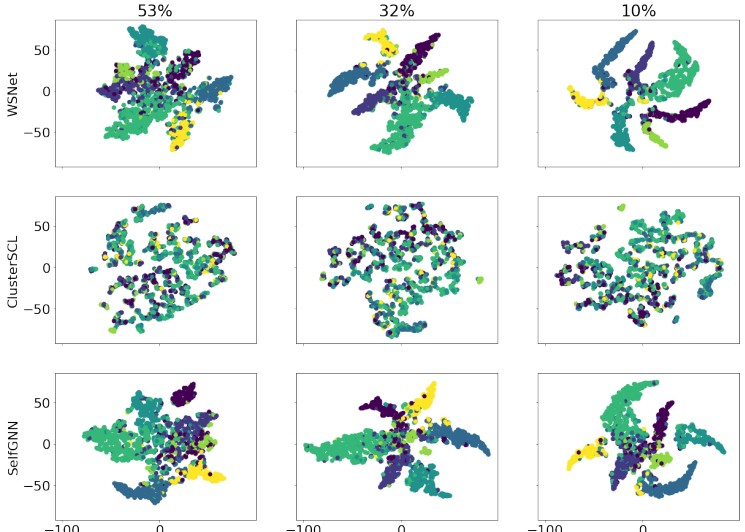

Figure 2: TSNE representations of predicted probability vectors under different levels of noise on Cora dataset. Colors indicate class labels. We observe better class separability for WSNET across different noise levels especially compared to ClusterSCL.

