# OpenReview forum: "Weakly Supervised Graph Contrastive Learning"
_ICLR.cc/2024/Conference — Submitted to ICLR 2024_

### Official Review · Reviewer_YxoE · 2023-10-28

**Soundness:** 2 fair
**Presentation:** 3 good
**Contribution:** 2 fair
**Rating:** 5
**Confidence:** 3

**Summary:**

This paper targets the weak/noisy label contrastive learning task. The contributions lie in two perspectives: Under the context of weak label  graph representation learning 1) the authors demonstrate that prior graph contrastive learning works do not show obvious robustness across different levels of noise; 2) the authors propose WSNet, which shows relatively superior robustness over weak/noisy labels. The authors also conduct ablation experiments to prove the necessity of the combination of the two defined losses.

**Strengths:**

1.	The paper is well-written. It is very easy to follow, and the authors provide thorough details about the problem setting, loss definition, as well as experimental settings.
2.	The experiments are extensive under the weak/noisy label setting.
3. The authors provide codes for reproduction.

**Weaknesses:**

1. The weak/noisy label setting appears to be confined to a limited context, especially on graphs. While I acknowledge the contribution of this paper in this specific area, its applicable generality to real-world graph datasets is questionable.
2. The analysis to the baseline GCL methods performance under noisy settings are shallow to some extent. The authors may consider some further analysis. For example, how well would each baseline perform under different types of classifier? What are the samples in common that are "robust" to such weak/noisy labels for each baseline?
3. An MLP is widely-used classifier as well. The authors may consider adding it to the experiments.

**Questions:**

Refer to weaknesses.

---

> ### Author Response · Authors · 2023-11-17
> **Response to reviewer YxoE**
>
> We thank the reviewer for their feedback and suggestions.
>
> 1. *The weak/noisy label setting appears to be confined to a limited context, especially on graphs. While I acknowledge the contribution of this paper in this specific area, its applicable generality to real-world graph datasets is questionable.*
>
> Response: We believe that such a weak supervision approach will benefit graph learning especially in text-attributed graphs where it is easier to obtain weak labels based on keyword matching, etc. Even with the recently growing popularity of text-attributed graphs such as in recommender systems[2] and fake news detection[1], where obtaining ground truth labels is challenging, the proposed weak supervision approach is useful.
>
> 2. *The analysis to the baseline GCL methods performance under noisy settings are shallow to some extent. The authors may consider some further analysis. For example, how well would each baseline perform under different types of classifier? What are the samples in common that are "robust" to such weak/noisy labels for each baseline?*
>
> Response: We agree that experimenting with different types of classifiers is interesting and we will consider it for our future work. In the current paper, since we are interested in studying the robustness of the learned GCL embeddings to label noise in downstream classification, we think that the choice of the classifier would have relatively less effect on the final conclusion as the same classifier is used for comparing all baseline methods. It might be challenging to identify the common samples that are robust to weak labels for each baseline. Do you mean to consider robustness to be based on low noise or based on correct prediction? Please clarify the question.
>
> 3. *An MLP is widely-used classifier as well. The authors may consider adding it to the experiments.*
>
> Response: Typically in contrastive learning, the learned embeddings are evaluated for linear separability using a simple linear classifier such as logistic regression. Although MLPs are generally better than LR, they further transform the learned node representations to another embedding space (if multiple layers). Having said that, it is an experiment that can be added but we believe tells us little more about the effectiveness of our method.
>
> [1] Adrien Benamira, Benjamin Devillers, Etienne Lesot, Ayush K Ray, Manal Saadi, and Fragkiskos D Malliaros. 2019. Semi-supervised learning and graph neural networks for fake news detection. In Proceedings of the 2019 IEEE/ACM International Conference on Advances in Social Networks Analysis and Mining. 568–569.
>
> [2] Jason Zhu, Yanling Cui,Yuming Liu, Hao Sun, Xue Li, Markus Pelger, Tianqi Yang, Liangjie Zhang, Ruofei Zhang, and Huasha Zhao. 2021. Textgnn:Improving text encoder via graph neural network in sponsored search. In Proceedings of the Web Conference 2021.2848–2857.

---

> > ### Comment · Reviewer_YxoE · 2023-11-22
> > **Response to the Authors**
> >
> > Thank you for the response. I agree that the type of the classifier may not effect much on the conclusion. However, there are still some other ways to validate the qualities of the embeddings (under different levels of noise). For example, show TSNE under different levels and compare it with other baselines (this is a shallow sample. the authors may consider others). I apologize if I am not clear enough. By "the samples in common that are 'robust' to such weak/noisy labels for each baseline", I mean identifying the sample group that is robust to the noise. For example, nodes with high degrees. I would like to see deeper analysis to show why/how WSNet works, rather than simply showing that it works. I hope it clarifies your questions.

---

> > > ### Author Response · Authors · 2023-11-22
> > > **Response regarding embedding quality and working of WSNet**
> > >
> > > Thank you for your clarifications.
> > >
> > > Firstly, to address your comment about visually evaluating the quality of the embeddings, as per your suggestion, we plotted TSNE representations of the embeddings learned using WSNet and some baselines. We included this plot in the revised version of the paper in Figure 1, Appendix D. We note that the embeddings learned using WSNet are more separated with respect to class labels compared to baselines (ClusterSCL also uses weak label signals in the contrastive learning process through $L_{SupCon}$ whereas other baselines do not take weak labels into account while learning embeddings). Despite being affected by the noise in the weak labels, we find that WSNet embeddings are better aligned with the class labels.
> > >
> > > Secondly, we used the learned embeddings to train a linear classifier (LR) with different levels of noisy labels and plot their TSNE representations in Figure 2, Appendix D. Again, we observe that WSNet embeddings are better aligned with class labels especially compared to ClusterSCL.
> > >
> > > Thirdly, as we understood it, your last comment was to analyze the nodes in the graph whose predicted label matches with its true label despite being trained using the noisy weak labels to identify commonalities/properties. This is an interesting suggestion. For the sake of simplicity, we considered the nodes robust to noise that were common between our method and ClusterSCL in Cora dataset. We observed that of the 79 common robust nodes out of 998 test nodes, 44% belonged to the same graph community. We did not observe any relation with graph properties such as high degree, etc. Having said that, we believe that the reasons WSNet works well is:
> > > 1. it leverages signals from the weak labels
> > > 2. it leverages the graph structure explicitly while sampling positive pairs, which acts as a sort of *regularizer* for the label noise.
> > >
> > > As a result of the above two, WSNet does better than a) other supervised GCL methods that do not explicitly use the graph structure in the sampling process and b) other GCL methods that do not leverage signals from the labels.
> > >
> > > We hope that this answers your question and request you to kindly go through the revised version of the paper. We are happy to engage in more discussion or answer any questions in the remaining time.

---

### Official Review · Reviewer_waBU · 2023-10-29

**Soundness:** 2 fair
**Presentation:** 2 fair
**Contribution:** 2 fair
**Rating:** 6
**Confidence:** 4

**Summary:**

This paper proposes a noisy-label learning method for graph contrastive learning which incorporates signals from weak labels. The  authors aim to explore the robustness of GCL methods to label noise and combine weak labels with graph communities to obtain better node representations. Extensive experiments illustrate the robustness of the node representations learned using GCL to weakly supervised classification and the effectveness of using weak labels to learn more robust embeddings.

**Strengths:**

1.  The description and formulation of the proposed algorithm are clear. The idea of combining the weak labels with graph communities to learn node representations  is novel.

2. The experimental analysis is extensive and the article provides a comprehensive evaluation of the proposed algorithm on multiple benchmark datasets, demonstrating its effectiveness in various noise settings.

3. The authors have provided sufficient details about the datasets and the experimental setup, which is commendable.

**Weaknesses:**

1. How does the weak label be generated? The author only state that the weak label is generated by the labeling function but without no more elaboration. If the label is not given, then how the weak label can be generated with a certain accuracy? A comparison on different labeling function could also be helpful.

2. The definition of robustness of GCL to label noise is very confusing. 1) Is label noise meaning the inaccuracy of true label or generated weak label? 2) It follows a logical intution that when there is a high level of lable noise in a dataset, the accuracy of a model trained on this dataset is likely to be low. If the label noise in a dataset as high as 53%, ahieveing high accuracy could be somewhat meaningless.

3. The idea of this paper is very similar to cluster-based graph contrastive learning such as [1] which utilize the cluster or community information as auxiliary information for learning objective.  Thus, the author should concentrate on comparining with these baselines.

4. The methods should be evaluated on more large-scale datasets such as ogbn datasets or Aminer-CS datasets. The datasets containing only hundreds or thousands of nodes are less convincing.

5. The presentation of the paper should be improved. For example, the caption of table should apperaove above the table; It is better not to place any context between two tables.
[1] CLEAR: Cluster-Enhanced Contrast for Self-Supervised Graph Representation Learning

**Questions:**

See weakness above.

---

> ### Author Response · Authors · 2023-11-17
> **Response to reviewer waBU**
>
> We thank the reviewer for their feedback and suggestions.
>
> 1. *How does the weak label be generated? The author only state that the weak label is generated by the labeling function but without no more elaboration. If the label is not given, then how the weak label can be generated with a certain accuracy? A comparison on different labeling function could also be helpful.*
>
> Response: In our paper, the weak labels are generated using Algorithm 3 which requires the ground truth label as it is a synthetic experiment setup. In real-world problems, when the ground truth is not available, LFs look for clues within the data. For example, in citation networks like Cora, the abstract of the papers (nodes) contains certain keywords related to a class label. So if these keywords are present in a paper abstract, we can assign the corresponding class as its weak label.
>
> 2. *The definition of robustness of GCL to label noise is very confusing. 1) Is label noise meaning the inaccuracy of true label or generated weak label? 2) It follows a logical intution that when there is a high level of lable noise in a dataset, the accuracy of a model trained on this dataset is likely to be low. If the label noise in a dataset as high as 53%, ahieveing high accuracy could be somewhat meaningless.*
>
> Response: To clarify, the weak label or label noise is only on the generated weak label and not the ground truth label. The proposed WSNet model is trained using the noisy labels but evaluated with the ground truth. Despite training the model with 50% noisy labels, it achieves a 68% F1 score (on Cora) when the predicted labels are compared to the ground truth labels (which have no noise).  In other words, the train labels are always noisy and the test labels used for evaluating the predictions are clean.
>
> 3. *The idea of this paper is very similar to cluster-based graph contrastive learning such as [1] which utilize the cluster or community information as auxiliary information for learning objective. Thus, the author should concentrate on comparining with these baselines.*
>
> Response: Thank you for sharing this reference. This work is focused on graph prediction whereas we are focused on node-level tasks. However, we will try our best to compare WSNet with this method for the future versions of our paper due to the time constraint and inaccessibility of their code.
>
> 5. *The presentation of the paper should be improved. For example, the caption of table should apperaove above the table; It is better not to place any context between two tables.*
>
> Response: Thank you for the suggestion. We will update this in the revised version.

---

> > ### Comment · Reviewer_waBU · 2023-11-20
> >
> > Thank you to the authors for their response to my initial review. Despite the clarifications provided, my primary concern regarding the novelty of this work, especially in relation to existing literature, remains unaddressed:
> >
> > 1. Comparison with Cluster or Community-Based Methods: The concept presented in the paper closely aligns with cluster or community-based graph contrastive learning approaches. These methods categorize nodes into different clusters/communities and utilize these auxiliary information in contrastive learning, often performing "contrastive learning" within the same community/cluster. A search  keywords including “cluster-based” and “graph contrastive learning” yields [1]. The authors need to conduct a more comprehensive survey of these methods and draw clear distinctions, highlighting the unique advantages of their approach. Even if some of these methods are not open-sourced, it’s essential to articulate what sets this work apart from such cluster/community-based strategies.
> >
> > 2. Application of CV Domain Methods to Graphs: The paper seems to transpose contrastive learning techniques from the CV domain, as seen in [2], to graph data. However, [2] is only briefly mentioned in the related work section, without a detailed comparison. The authors should thoroughly compare their approach with [2], clarifying which aspects are specifically designed for graphs. If the methodology primarily involves substituting CV encoders with GNNs and replacing image clustering with node clustering methods, it might limites the novelty of the paper.
> >
> > In conclusion, a more extensive literature survey is necessary, comparing the proposed method with similar cluster-based graph contrastive learning approaches and the foundational weakly supervised contrastive learning method in [2]. The authors should intuitively explain their method’s advantages at the very least on a methodological level. Therefore, I maintain my original score for this paper.
> >
> > [2] Weakly supervised contrastive learning. ICCV 2021.

---

> > > ### Author Response · Authors · 2023-11-20
> > > **Clarification regarding related works**
> > >
> > > We thank the reviewer for their clarifying comments and questions and try our best to address them below.
> > >
> > > 1. We agree that the clusters/communities have been utilized previously as auxiliary information for contrastive learning as in [1]. However, our work is different in three aspects. Firstly, while both [1] and our method rely on obtaining structural information from graph clusters, the motivation and end-goal for doing so in the two are different. We use graph communities to find densely connected subgraphs based on minimizing modularity with the aim of finding sets of nodes with similar structural properties. On the other hand, [1] appears to be finding graph partitions in order to capture the underlying semantics within local substructures to learn overall better graph embeddings. Secondly, we rely on graph communities for sampling positive pairs of nodes for contrastive learning, i.e, nodes within the same community should be closer to each other in the embedding space whereas in [1], among other things, they treat different partitions/clusters as two different positive views and contrast them with each other. Thirdly and most importantly, ours is a weakly supervised contrastive learning method where we also leverage the signals from the weak class labels to improve the contrastive learning, differentiating our work from [2]. Thank you for this comment and in the revised version we will include this paper[1] and discuss how our work is different from it.
> > >
> > > 2. Our work is inspired by the SupCon[3] contrastive loss from the CV domain which has also been used in the graph domain [4] (ClusterSCL with which we compare our work). While it is also related to [2], we highlight that the weak labels used in [2] are inherently different from ours. [2] assigns similar nodes within the same connected components the same weak label which are different from our noisy/weak class labels which are assumed to be provided along with the input graph. Moreover, the optimized contrastive loss in [2] consists of 4 parts of which 2 are dependent on image crops and cannot directly be applied to graphs. The remaining two are $L_{SupCon}$ and another component based on augmented views. Our baseline titled SupCon is essentially a combination of these two loss components but defined in [4] for the graph domain. We hope this clarifies that our method is not the same as [2] with CV components substituted with graph components. We will clarify this better in the revised version.
> > >
> > > Thank you for both the comments and we will include a more detailed discussion regarding these two points and related literature in the revised version. We hope we have answered your questions and are happy to discuss more.
> > >
> > >
> > > [3] Prannay Khosla, Piotr Teterwak, Chen Wang, Aaron Sarna, Yonglong Tian, Phillip Isola, Aaron Maschinot, Ce Liu, and Dilip Krishnan. Supervised contrastive learning. arXiv preprint arXiv:2004.11362, 2020
> > >
> > > [4] Wang, Yanling, et al. "ClusterSCL: cluster-aware supervised contrastive learning on graphs." Proceedings of the ACM Web Conference 2022. 2022.

---

### Official Review · Reviewer_oC5W · 2023-10-31

**Soundness:** 2 fair
**Presentation:** 2 fair
**Contribution:** 2 fair
**Rating:** 5
**Confidence:** 3

**Summary:**

This paper investigates node representation learning in Graph Contrastive Learning (GCL) under weak supervision. Firstly, the paper analyzes the robustness of node representations learned by existing methods under weak supervision and concludes that they are all affected by label noise. To mitigate this issue, this work leverages graph structures to identify more relevant positive sample pairs. Specifically, it identifies nodes belonging to the same community from the entire graph as positive sample pairs.

Overall, I think the paper is good but not quite up to ICLR standards. Mainly, it lacks some necessary experiments, such as efficiency analysis, feasibility on large-scale graphs, and so on. Further analysis of the possible impacts of the work would be helpful.

**Strengths:**

1. The experimental performance is good.

2. The work conducts experiments on different levels of homogenous graph datasets, constructing positive sample pairs by selecting nodes belonging to the same community. I believe this is meaningful for graph contrastive learning.

**Weaknesses:**

1. Identifying nodes belonging to the same community from the entire graph is a computationally intensive operation. Including an analysis of efficiency would enhance the completeness of this work.

2. The paper lacks diagrams/figures, training details, or efficiency analysis.

3. Theoretical analysis in this paper is insufficient.

4. The dataset used in this work is relatively small. The authors mention in the paper that this work can be extended to larger datasets. Conducting experiments on larger datasets,  such as the OGB dataset, would further demonstrate the effectiveness of the work. Additionally, analyzing the feasibility of identifying positive sample pairs on large graphs with acceptable efficiency is worth considering.

5. The paper proposes a way to improve the learning of node embeddings learned with graph contrastive under weak supervision. The paper also provides some experiments that show that their model can achieve good performance. However, the experiments are not sufficient, and the innovativeness is not up to ICLR standards.

**Questions:**

1. Which labeling functions (LFs) were used to derive the weak label matrix, Lambda? As far as I remember, the famous citation network triplet (Cora, Citeseer, and Pubmed) does not come with such weak labels. Furthermore, in the context of Majority Voting, how is a tie among LFs resolved?
2. The Louvain algorithm [1] identifies exclusive communities, implying each node associates with only one community. This exclusivity is often incongruent with real-world scenarios. For instance, individuals in social networks typically affiliate with multiple groups, such as family, friends, and colleagues. Similarly, in biological contexts, genes or proteins often participate in multiple pathways. Furthermore, the Louvain method can yield poorly connected communities [2]. Could you elucidate further on the algorithm's application, such as the number of communities detected versus class count and how effective it helps with “finding nodes with a similar graph structure”? Why not opt for other superior methods, such as the Leiden algorithm?
3. Hard samples, particularly hard negatives, are pivotal for representation learning under the Contrastive framework [3]. The Supervised Contrastive Loss study [4] further emphasizes the significance of hard samples over easy ones. Could you provide further insight into L-supcon, given its centrality in your method?
4. The proposed loss combines Self-Supervised Contrastive Loss L-s and Supervised Contrastive Loss L-supcon. This combination suggests equal influence from both losses, yet intuitively, L-supcon seems more potent than L-s. Do you think it is necessary to account for their respective contributions to representation learning, perhaps by introducing and searching for a hyperparameter?
5. In PI-GNN [5], the authors employ noise ratios of 0.0, 0.2, 0.4, 0.6, and 0.8, which appear more intuitive. Despite PI-GNN focusing on image datasets and WSNET on graph datasets, The chosen noise ratios in this study (High 53%, Medium 32%, Low 10%) are notably specific. Could you elucidate the rationale behind these values and explain your approach to introducing noise to the original labels?
6. How do you account for the enhanced performance on non-homophilous graphs? Might this improvement be ascribed to the community detection algorithm?
7. In the paper, it is mentioned that weak labels have accuracies set at 47\%, 68\%, and 90\%. I am curious about how these weak labels are generated and how their accuracies are controlled.

**Typos, Formatting Issues, and Grammatical Errors:**
1. In abstract sentence 3, add a comma after “instead”, before “particularly” in sentence 10, and before “and” in sentence 11.
2. In Section 1, paragraph 3, sentence 7, add a comma before “or”, and “citations” should be in its singular form to align with “networks”.
3. In section 1, paragraph 5, sentence 5, “are” should be “is” to agree with “answering these questions”.
4. In section 2, paragraph PWS, sentence 3, add a comma before “and”.
5. In section 2, paragraph PWS, sentence 6, “straight-forward” should be “straightforward”.
6. In section 2, paragraph PWS, sentence 8, “to study” should be “on studying” to align with “on weak label aggregation”.
7. In section 2, paragraph NLL, sentence 1, “straight-forward” should be “straightforward”.
8. In section 2, paragraph NLL, sentence 3, “Most” should be “The most”.
9. In Section 4, Paragraph 1, in sentence “Here, we sample a node’s positive from the set of nodes that has it’s same aggregated label and negatives from the remaining nodes”, “it’s” seems redundant and erroneous.

**References**\
[1] Blondel, Vincent D; Guillaume, Jean-Loup; Lambiotte, Renaud; Lefebvre, Etienne (9 October 2008). Fast unfolding of communities in large networks. Journal of Statistical Mechanics: Theory and Experiment. 2008 (10): P10008.\
[2] Traag, V.A., Waltman, L. & van Eck, N.J. From Louvain to Leiden: guaranteeing well-connected communities. Sci Rep 9, 5233 (2019).\
[3] Khosla, P., Teterwak, P., Wang, C., Sarna, A., Tian, Y., Isola, P., Maschinot, A., Liu, C., & Krishnan, D. (2020). Supervised Contrastive Learning.\
[4] Kalantidis, Y., Sariyildiz, M.B., Pion, N., Weinzaepfel, P., & Larlus, D. (2020). Hard Negative Mixing for Contrastive Learning. \
[5] Du, X., Bian, T., Rong, Y., Han, B., Liu, T., Xu, T., Huang, W., Li, Y., & Huang, J. (2021). Noise-robust Graph Learning by Estimating and Leveraging Pairwise Interactions.\
[6] "Grammarly." Wikipedia, Wikimedia Foundation, 27 September 2023, en.wikipedia.org/wiki/Grammarly.

---

> ### Author Response · Authors · 2023-11-17
> **Response to reviewer oC5W**
>
> We thank the reviewer for their feedback and suggestions.
>
> 1. *Identifying nodes belonging to the same community from the entire graph is a computationally intensive operation. Including an analysis of efficiency would enhance the completeness of this work.*
>
> Response: This is true that there is a computational overhead, however, this can be implemented such that it only incurs a one-time computation before the model training. The model training itself is pretty fast. Specifically, we need to first find the communities. Community detection is a one-time computation that happens before the model training. Since we use the Louvain algorithm for detecting communities, there is an overhead of O($nlogn$) for graphs with $n$ nodes. Next, we identify and sample all the positives and negatives we need during the training based on the communities, and the similarities of their weak label distributions. This has a runtime of  $O(np^2)$, where $p$ is the size of the largest community in the graph and $p << n$ in most real world graphs. We also report the runtime used for these pre-training computations as well as the actual training on Cora and Pubmed in Tables 1 and 2 using Louvain in the common response.
>
> 2. *The paper lacks diagrams/figures, training details, or efficiency analysis.*
>
> Response: We will add the training details and run-time analysis to the revised paper. All the experiments were run locally on an 8-cpu core Macbook M2 computer. WSNet was trained for 50 epochs and the results averaged over 5 runs is reported in the paper. For all the other GCL methods, we used the official code released by the authors of the respective papers and tried our best to use the recommended hyperparameters wherever relevant.
>
> 5. *The paper proposes a way to improve the learning of node embeddings learned with graph contrastive under weak supervision. The paper also provides some experiments that show that their model can achieve good performance. However, the experiments are not sufficient, and the innovativeness is not up to ICLR standards.*
>
> Response: We would like to highlight that the main novelty of our work is to introduce weakly supervised graph contrastive learning. There are no other works, to the best of our knowledge, that do GCL with weak labels. We also believe that weakly supervised graph learning is an important field of study for solving graph-based real world problems such as in recommender systems or misinformation detection where only weak labels may be available. Secondly, we systematically study existing GCL methods in the framework of weak supervision which is novel and our results show that there is scope for further research in this direction.
>
> Q1. *Which labeling functions (LFs) were used to derive the weak label matrix, Lambda? As far as I remember, the famous citation network triplet (Cora, Citeseer, and Pubmed) does not come with such weak labels. Furthermore, in the context of Majority Voting, how is a tie among LFs resolved?*
>
> Response: The LFs were synthetically generated as per Algorithm 3. For a given accuracy/coverage, we flip the labels (given the ground truth) to get weak labels. We repeat this process $m$ separate times to obtain $m$ weak labels for each node. In case of majority voting tie, one of the tied labels is selected uniformly at random.

---

> ### Author Response · Authors · 2023-11-17
> **Response to reviewer oC5W (contd.)**
>
> Q2. *The Louvain algorithm [1] identifies exclusive communities, implying each node associates with only one community. This exclusivity is often incongruent with real-world scenarios. For instance, individuals in social networks typically affiliate with multiple groups, such as family, friends, and colleagues. Similarly, in biological contexts, genes or proteins often participate in multiple pathways. Furthermore, the Louvain method can yield poorly connected communities [2]. Could you elucidate further on the algorithm's application, such as the number of communities detected versus class count and how effective it helps with “finding nodes with a similar graph structure”? Why not opt for other superior methods, such as the Leiden algorithm?*
>
> Response: Thank you for your suggestion. We acknowledge that Leiden is a good alternative to Louvain especially for scalability and we will add this discussion to the revised version of the paper. However, we would also like to highlight that our proposed method is not dependent on the community detection algorithm. Our contribution is more on the idea of sampling positives from detected communities for robustness to weak labels. We also experimented with using Leiden instead of Louvain and reported the results in Tables 1 (Cora) and 2 (Pubmed) in the common response. We observe that there is not much difference in the number of detected communities and mean entropy of class labels within communities for both the methods. We do note that Leiden has a slightly higher NMI and ARI score with the ground-truth labels compared to Louvain which could be the reason why WSNet performs slightly better using Leiden on Cora. The NMI and ARI between the detected community labels of the two methods is also high for Cora but quite low for Pubmed.
> “Could you elucidate further on the algorithm's application, such as the number of communities detected versus class count and how effective it helps with “finding nodes with a similar graph structure”?” Do you mean to ask if matching the number of communities and number of classes would result in better performance? Clarification on this question will be helpful.
>
> Q3. *Hard samples, particularly hard negatives, are pivotal for representation learning under the Contrastive framework [3]. The Supervised Contrastive Loss study [4] further emphasizes the significance of hard samples over easy ones. Could you provide further insight into L-supcon, given its centrality in your method?*
>
> Response: Thanks for bringing up this point. We agree that hard negatives are important for contrastive learning and this is an interesting future work for consideration. As our work is a preliminary study of weakly supervised GCL, we simply sample positives and negatives based on the  *aggregated weak labels*. Nodes from different classes are considered negatives and nodes from the same class are positives, similar to the original L_{SupCon} loss.
>
> Q4. *The proposed loss combines Self-Supervised Contrastive Loss L-s and Supervised Contrastive Loss L-supcon. This combination suggests equal influence from both losses, yet intuitively, L-supcon seems more potent than L-s. Do you think it is necessary to account for their respective contributions to representation learning, perhaps by introducing and searching for a hyperparameter?*
>
> Response: We actually had experimented with this idea and did not see improvement, in particular in the high noise setting. When the quality of labels is low, weighting $L_{SupCon}$ higher, negatively impacts the overall performance. This is expected as $L_{SupCon} relies entirely on the weak labels for sampling positives and negatives. Here we report the performance of WSNet with (40, 60), (30, 70) and (20, 80).
> | Cora                   | 53%              | 32%              | 10%              |
> | ---------------------- | ---------------- | ---------------- | ---------------- |
> | L_S + L_{SupCon}       | 0.68 $\\pm$ 0.02 | 0.80 $\\pm$ 0.04 | 0.92 $\\pm$ 0.02 |
> | 0.4L_S + 0.6L_{SupCon} | 0.41 $\\pm$ 0.04 | 0.79 $\\pm$ 0.07 | 0.93 $\\pm$ 0.04 |
> | 0.3L_S + 0.7L_{SupCon} | 0.43 $\\pm$ 0.03 | 0.73 $\\pm$ 0.05 | 0.92 $\\pm$ 0.04 |
> | 0.2L_S + 0.8L_{SupCon} | 0.52 $\\pm$ 0.03 | 0.76 $\\pm$ 0.06 | 0.92 $\\pm$ 0.04 |

---

> ### Author Response · Authors · 2023-11-17
> **Response to reviewer oC5W (contd.)**
>
> Q5. *In PI-GNN [5], the authors employ noise ratios of 0.0, 0.2, 0.4, 0.6, and 0.8, which appear more intuitive. Despite PI-GNN focusing on image datasets and WSNET on graph datasets, The chosen noise ratios in this study (High 53%, Medium 32%, Low 10%) are notably specific. Could you elucidate the rationale behind these values and explain your approach to introducing noise to the original labels?*
>
> Response: Since we are interested in the setting where each node is associated with multiple weak labels we varied both the number of LFs ($m$) and individual LF accuracies ($p_a$) pairs as ($m$, $p_a$) =  [(5, 0.45), (10, 0.65), (50, 0.55)]. These weak labels are aggregated using majority vote to obtain, 53%, 32% and 10% accuracies. In other words, we only control the $m$ and $p_a$ parameters and not the final accuracy/noise of the aggregated labels. We will highlight this better in the revised version of the paper.
>
> Q6. *How do you account for the enhanced performance on non-homophilous graphs? Might this improvement be ascribed to the community detection algorithm?*
>
> Response: Yes, we believe the improved performance on non-homophilous graphs may be attributed to positive sampling from communities. We will highlight this in the revised version.
>
> Q7: *In the paper, it is mentioned that weak labels have accuracies set at 47%, 68%, and 90%. I am curious about how these weak labels are generated and how their accuracies are controlled.*
>
> Response: Individual $m$ weak labels are generated using Algorithm 3 and are then aggregated using majority vote. We have clarified this in the revised version.
>
> The grammatical errors and typos have been noted with thanks and corrected in the revised version.

---

> > ### Comment · Reviewer_oC5W · 2023-11-22
> >
> > Thank you for the detailed reply. The authors have resolved most of the concerns and I will increase the score to 5.

---

> > > ### Author Response · Authors · 2023-11-22
> > > **Thank you for the reply**
> > >
> > > We sincerely thank you for your discussion. If you have any further questions or concerns, we are happy to continue the discussion in the remaining time.

---

### Official Review · Reviewer_G4h6 · 2023-11-02

**Soundness:** 2 fair
**Presentation:** 2 fair
**Contribution:** 2 fair
**Rating:** 6
**Confidence:** 3

**Summary:**

This paper proposes a novel graph contrastive learning method, named WSNET, to learn node representations when weak or noisy labels are present. The authors conducted experiments to compare the robustness of current GCL node representation methods under weak supervision and found that incorporating weak label information into contrastive learning using WSNET can improve node representation learning.

**Strengths:**

The authors propose a novel approach to improve the node representation learning ability of GCL when there is noise in the node labels.

**Weaknesses:**

1. The authors spend too much space on Research Question 1, which evaluates the robustness of existing GCL methods. This is only a process of running baselines and cannot be the main innovation and contribution of the paper. The authors should focus more on the proposed method.
2. Even with the amount of space devoted to RQ1, I don't think the authors' analysis is deep enough. For example, the authors conclude from the experimental results that baselines that use neighborhood-based sampling of positive pairs perform better than others in the high noise setting. What is the reason for this, and the authors should provide some analysis.
3. The notations are inconsistent, which makes it difficult for readers to understand the algorithm. For example, in the first line of Algorithm 1, $\lambda_i$ looks like a scalar, but Section 3 says that $\lambda$ is a label function. Additionally, are $\Lambda_i$ and $\Lambda[V_i]$ representing the same vector?
4. Will the algorithm's results differ significantly depending on the community search algorithm used and the label aggregation method used? The authors should provide more experimental results to compare the possibilities.
5. Will the size of the negative samples $r$ have a significant impact on the results? The authors should provide a sensitivity analysis of the parameters.
6. I believe that the authors should introduce larger-scale datasets, such as OGB data, as the current experimental datasets cannot verify the effectiveness of the proposed method in real-world situations.

**Questions:**

1. It is unclear why only the node with the highest similarity is sampled as a positive sample for the $L_S$ loss. It would be interesting to investigate whether using a top-k sampling strategy would be feasible and potentially improve the results.
2. It is unclear why WSNET performs better in the High Noise setting than in the Medium Noise setting on the Texas dataset.

---

> ### Author Response · Authors · 2023-11-17
> **Response to reviewer G4h6**
>
> We thank the reviewer for their insightful comments.
>
> 1.  *The authors spend too much space on Research Question 1, which evaluates the robustness of existing GCL methods. This is only a process of running baselines and cannot be the main innovation and contribution of the paper. The authors should focus more on the proposed method.*
>
> Response: We would like to clarify that although RQ1 is not the main innovation point in the paper, there is still value in studying how different GCL methods perform in the presence of label noise as this comparative study has not been done before. We clarify that these experiments are not only baselines but also constitute the first study on the robustness of GCL methods to weak labels. Moreover, we observe that no single method consistently outperforms others showing that there is scope for further research in this direction. We will discuss this better in the revised version of the paper.
>
> 2. *Even with the amount of space devoted to RQ1, I don't think the authors' analysis is deep enough. For example, the authors conclude from the experimental results that baselines that use neighborhood-based sampling of positive pairs perform better than others in the high noise setting. What is the reason for this, and the authors should provide some analysis.*
>
> Response: GNNs typically implement neighborhood aggregation for learning efficient and robust node representations and this supports our observations that neighborhood-based sampling makes the embeddings more robust to weak labels compared to node augmentations. On contrasting views of the same nodes versus neighboring nodes, the learned embeddings contain more information about the neighbors in the latter than the former. We will expand on this discussion in the revised version.
>
> 3. *The notations are inconsistent, which makes it difficult for readers to understand the algorithm. For example, in the first line of Algorithm 1, $\lambda_i$ looks like a scalar, but Section 3 says that \lambda is a label function. Additionally, are $\Lambda_i$ and $\Lambda[V_i]$ representing the same vector?*
>
> Response: Thank you for pointing these out. We will clarify these notations in the revised version of the paper. Mainly that $\lambda_i$ is not a scalar but the output vector obtained by applying labeling functions $\lambda$ on $i$th node and $\Lambda_i$ and $\Lambda[V_i]$ do represent the same vector.
>
> 4. *Will the algorithm's results differ significantly depending on the community search algorithm used and the label aggregation method used? The authors should provide more experimental results to compare the possibilities.*
>
> Response: Both the community search algorithm and the label aggregation methods are modules that can be replaced in our proposed WSNet framework. Our method is not tied to these modules. For our experiments, we chose the simplest and most popular method in both cases.
>
>   a) More specifically, we used the simple majority vote label aggregator to demonstrate the effectiveness of our proposed WSNet. On using better label aggregation methods the accuracy of the aggregated labels would increase resulting in the improvement of the overall performance of WSNet. In our work we show that even using a simple majority vote label aggregation, WSNet outperforms other baselines. We repeated our experiment using the Snorkel[1] label aggregator and reported the results in Table below. We observe that the Snorkel accuracy of the aggregated labels is lower than MV for all noise levels and as a result WSNet trained using the corresponding labels has a lower performance than MV.
> | Cora                     | 53%              | 32%              | 10%              |
> | ------------------------ | ---------------- | ---------------- | ---------------- |
> | MV accuracy              | 0.47             | 0.68             | 0.90             |
> | WSNET with Majority Vote | 0.68 $\\pm$ 0.02 | 0.80 $\\pm$ 0.04 | 0.92 $\\pm$ 0.02 |
> | Snorkel accuracy         | 0.43             | 0.69             | 0.84             |
> | WSNET with Snorkel       | 0.39 $\\pm$ 0.04 | 0.73 $\\pm$ 0.07 | 0.86 $\\pm$ 0.09 |
>
>  b) Similarly, for community detection, we address the comments in the common response.

---

> ### Author Response · Authors · 2023-11-17
> **Response to reviewer G4h6 (contd.)**
>
> 5. *Will the size of the negative samples r have a significant impact on the results? The authors should provide a sensitivity analysis of the parameters.*
>
> Response: Generally, the higher the value of $r$, the better the performance. However, with higher values of $r$, the computational complexity also increases as the denominator term in the $L_S$ (Eq 3 in the paper). We ran experiments on the Cora dataset for varying values of $r$ and reported the results below. We will add these results and discussion to the appendix of the revised paper due to space constraints. We observe a similar performance for $r=10$ and $r=20$ with slight increase in runtime. Given that the performance is similar, we used $r=10$ as in our experiments in the paper.
> | Cora     | 53%              | 32%              | 10%              | Runtime |
> | -------- | ---------------- | ---------------- | ---------------- | ------- |
> | $r = 5$  | 0.40 $\\pm$ 0.04 | 0.74 $\\pm$ 0.07 | 0.93 $\\pm$ 0.05 | 25.4    |
> | $r = 10$ | 0.68 $\\pm$ 0.02 | 0.80 $\\pm$ 0.04 | 0.92 $\\pm$ 0.02 | 28.3    |
> | $r = 20$ | 0.68 $\\pm$ 0.04 | 0.80 $\\pm$ 0.06 | 0.93 $\\pm$ 0.07 | 28.6    |
>
> Q1. *It is unclear why only the node with the highest similarity is sampled as a positive sample for the L_S loss. It would be interesting to investigate whether using a top-k sampling strategy would be feasible and potentially improve the results.*
>
> Response: We used only one positive sample keeping with the norm in contrastive learning. That is the standard for InfoNCE loss because it is considered to be similar to cross-entropy loss for classifying the correct positive sample from the remaining negative ones. Additionally, since the labels are noisy, by considering the top-k samples with similar weak label distribution, there is a higher chance of the noisy labels affecting the embeddings. Moreover, this can also become an issue for communities with only 1 node. However, we think this is an interesting setup to investigate. We ran an experiment to see if top-5 improves the results. As expected, for the higher noise settings, $k=1$ does better.
> | Cora  | 53%              | 32%              | 10%              | Runtime |
> | ----- | ---------------- | ---------------- | ---------------- | ------- |
> | $k=1$ | 0.68 $\\pm$ 0.02 | 0.80 $\\pm$ 0.04 | 0.92 $\\pm$ 0.02 | 28.3    |
> | $k=5$ | 0.53 $\\pm$ 0.05 | 0.75 $\\pm$ 0.07 | 0.94 $\\pm$ 0.03 | 29.5    |
>
> Q2. *It is unclear why WSNET performs better in the High Noise setting than in the Medium Noise setting on the Texas dataset.*
>
> Response: Considering that Texas is the smallest dataset we have with only 183 nodes, we observe a high variance in the results and although on average it seems the performance on high noise setting is better compared to medium noise setting, considering the variance, there is overlap in the ranges. This is also only observed with linear regression and not the random forest classifier, which might highlight this is partly due to simplicity of the downstream logistic regression classifier, more than the quality of the embeddings from WSNet. WSNet + RF shows the expected increase in performance with decrease in noise. We will expand the discussion in the paper on this.
>
> [1] Ratner A, Bach SH, Ehrenberg H, Fries J, Wu S, Ré C. Snorkel: Rapid Training Data Creation with Weak Supervision. Proceedings VLDB Endowment. 2017 Nov;11(3):269-282. doi: 10.14778/3157794.3157797. PMID: 29770249; PMCID: PMC5951191.

---

> > ### Comment · Reviewer_G4h6 · 2023-11-22
> >
> > Thank you for your response. I have no more questions, and I will raise my score to 6.

---

> > > ### Author Response · Authors · 2023-11-22
> > > **Thank you for the reply**
> > >
> > > Thank you for your comments and acknowledging our response. If you have any further questions or concerns, we are happy to continue the discussion in the remaining time.

---

### Author Response · Authors · 2023-11-17
**Common response to all reviewers**

The authors thank all the reviewers for their time, comments, suggestions and questions. We address a few of the common comments here and then respond individually to each of the reviewers.

### Comparing Louvain community detection with Leiden algorithm
Firstly, we would like to clarify that our method is not dependent on any one community detection method. Although we used Louvain, it can easily be replaced by any community detection algorithm. Our contribution is in showing that sampling positives from the same community as a given node helps with the downstream noisy node classification performance, especially in non-homophilous graphs. We experimented with using the Leiden community detection method instead of Louvain and reported the results in Table 1 (for Cora) and Table 2 (for Pubmed) below. We report the F1 classification score averaged over 5 runs for different noise levels. We don’t observe much difference in the detected communities as well as in the final classification performance. On Cora, Leiden has a slightly higher ARI with the ground truth resulting in slightly higher performance than using Louvain.

**Table 1:**
| Cora    | 53%              | 32%              | 10%              | Comm. Runtime | Sampling Runtime | Train time | Number of comms | Entropy of class labels within a comm | NMI (comm. label & class label) | ARI (comm. label & class label) |
| ------- | ---------------- | ---------------- | ---------------- | ------------- | ---------------- | ---------- | --------------- | ------------------------------------- | ------------------------------- | ------------------------------- |
| Leiden  | 0.69 $\\pm$ 0.07 | 0.88 $\\pm$ 0.05 | 0.93 $\\pm$ 0.02 | 0.07          | _                | _          | 84              | 0.17 $\\pm$ 0.34                      | 0.49                            | 0.30                            |
| Louvain | 0.68 $\\pm$ 0.02 | 0.80 $\\pm$ 0.04 | 0.92 $\\pm$ 0.02 | 0.59          | 15.2             | 29.5       | 81              | 0.17 $\\pm$ 0.34                      | 0.48                            | 0.29                            |


**Table 2:**
| Pubmed  | 53%               | 32%               | 10%               | Comm. detection Runtime | Sampling Runtime | Train time | Number of comms | Entropy of class labels within a comm. | NMI (comm. label & class label) | ARI (comm. label & class label) |
| ------- | ----------------- | ----------------- | ----------------- | ----------------------- | ---------------- | ---------- | --------------- | -------------------------------------- | ------------------------------- | ------------------------------- |
| Leiden  | 0.41 $\\pm$ 0.002 | 0.57 $\\pm$ 0.007 | 0.73 $\\pm$ 0.005 | 0.41                    | _                | _          | 37              | 0.69 $\\pm$ 0.27                       | 0.20                            | 0.10                            |
| Louvain | 0.41 $\\pm$ 0.00  | 0.59 $\\pm$ 0.01  | 0.70 $\\pm$ 0.01  | 0.87                    | 396.2            | 212.7      | 39              | 0.65 $\\pm$ 0.28                       | 0.21                            | 0.12                            |

---

> ### Author Response · Authors · 2023-11-18
> **Results on new large dataset - ogb-arXiv**
>
> We ran our method WSNet on a larger OGB dataset - ogbn-arXiv (169,343 nodes and 1,166,243 edges).
>
> The ogbn-arXiv dataset is a directed graph and it represents the citation network between all CS arXiv papers indexed by MAG [1].
> Each node is a paper and each directed edge indicates that one paper cites another one. Each paper is represented using a 128-dimensional feature vector obtained by averaging the skip-gram embeddings of words in its title and abstract. Each paper belongs to 1 of 40 CS subject areas so the task is a 40-class classification problem.
>
> We ran WSNet on arXiv and report the F1 classification score averaged across 5 runs for different noise levels. We also note the time taken in seconds for community detection, positive sampling and training. We note that 40-class classification is a much harder problem especially in a weak learning setup and consequently we observe low classification scores.
>
> |  | 53%               | 32%               | 10%               | Training | Community detection | Positive sampling |
> | --------- | ----------------- | ----------------- | ----------------- | -------- | ------------------- | ----------------- |
> | WSNet     | 0.10 $\\pm$ 0.002 | 0.19 $\\pm$ 0.003 | 0.32 $\\pm$ 0.004 | 1930.4   | 46.2                | 40886.3           |
>
> We tried running MVGRL, PI-GNN and Joint-Training baselines on arXiv but they all resulted in OOM and we let them run for over (1930.4 + 46.2 + 40886.3) seconds. We will try our best to run the remaining baselines for the revised version of the paper.
>
> [1] Kuansan Wang, Zhihong Shen, Chiyuan Huang, Chieh-Han Wu, Yuxiao Dong, and Anshul Kanakia. Microsoft academic graph: When experts are not enough. Quantitative Science Studies, 1(1):396–413, 2020.

---

### Author Response · Authors · 2023-11-21
**Summary of changes in the revised version**

We thank all the reviewers for their insightful comments, suggestions and questions. Taking these into account, we have made revisions to our original submission. The changes made in the paper are highlighted as blue text.

The summary of the changes made are below.

1. Better highlighted how our proposed work is different from related works that were pointed out by the reviewers.
2. Better highlighted the novelty of our work, particularly related to research question 1.
3. Better explained some of the conclusions of our experiments and results.
4. Included an experiment with varying the value of the hyperparamater $r$ which indicates the number of negative samples drawn for contrastive learning. These results are added in the appendix of the paper.
5. Included training details and a runtime complexity analysis, added to the appendix of the paper.
6. Made overall improvements in the paper presentation including correcting typos and grammatical errors pointed out by the reviewers.

We believe we have addressed most of the comments and are happy to answer any more questions the reviewers may have.

---

### Meta-Review · Area_Chair_436i · 2023-12-06

**Metareview:**

This paper introduces a graph contrastive learning approach that can handle weak labels in the training objective.

The reviewers find the suggested approach interesting and the experimental performance generally convincing. However, two caveats exist for both of those strengths: firstly, reviewer RwaBU remains unconvinced about novelty even after the rebuttal; this could be an issue about positioning the paper properly. Secondly, more than one reviewers comment on the small datasets used - the authors ran experiments on OGB however the analysis remains incomplete since results for competing methods are missing.

In addition to the discussion above, the key weakness of the paper is its presentation, which includes justifying design decisions, including informal as well as theoretical analyses. The majority of reviewers raise these points because they are somehow confused about various parts of the paper, as is evident by the many clarification questions asked. Even after the rebuttal (where the authors did a great job adding useful information), the authors still stayed with lower scores as it seems that major rewriting would be needed.

**Justification For Why Not Higher Score:**

Major rewriting needed even after post-rebuttal state.

**Justification For Why Not Lower Score:**

N/A

---

### Decision · Program_Chairs · 2024-01-16

Reject